# Daratumumab Interferes with Allogeneic Crossmatch Impacting Immunological Assessment in Solid Organ Transplantation

**DOI:** 10.3390/jcm11206059

**Published:** 2022-10-14

**Authors:** Chak-Sum Ho, Kyle R. Putnam, Christine R. Peiter, Walter F. Herczyk, John A. Gerlach, Yee Lu, Erica L. Campagnaro, Kenneth J. Woodside, Matthew F. Cusick

**Affiliations:** 1Gift of Hope Organ & Tissue Donor Network, Itasca, IL 60143, USA; 2Department of Medicine, College of Human Medicine, Michigan State University, East Lansing, MI 48824, USA; 3Gift of Life Michigan, Ann Arbor, MI 48108, USA; 4Biomedical Laboratory Diagnostics Program, College of Natural Science, Michigan State University, East Lansing, MI 48824, USA; 5Division of Nephrology, Department of Internal Medicine, University of Michigan Medicine, Ann Arbor, MI 48109, USA; 6Division of Hematology and Oncology, Department of Internal Medicine, University of Michigan Medicine, Ann Arbor, MI 48109, USA; 7Department of Pathology, University of Michigan Medicine, Ann Arbor, MI 48109, USA

**Keywords:** antibody-mediated rejection, crossmatch, daratumumab, end-stage renal disease, flow cytometry, human leukocyte antigen (HLA), multiple myeloma, transplantation

## Abstract

We report the first case of Daratumumab interference of allogeneic crossmatch tests repeatedly causing aberrant false-positive results, which inadvertently delayed transplant for a waitlisted renal patient with multiple myeloma. Daratumumab is an IgG1κ human monoclonal antibody commonly used to treat multiple myeloma, characterized by cancerous plasma cells and often leads to renal failure requiring kidney transplant, by depleting CD38-expressing plasma cells. In this case study, the patient had end-stage renal disease secondary to multiple myeloma and was continuously receiving Daratumumab infusions. The patient did not have any detectable antibodies to human leukocyte antigens but repeatedly had unexpected positive crossmatch by the flow cytometry-based method with 26 of the 27 potential deceased organ donors, implying donor-recipient immunological incompatibility. However, further review and analysis suggested that the positive crossmatches were likely false-positive as a result of interference from Daratumumab binding to donor cell surface CD38 as opposed to the presence of donor-specific antibodies. The observed intensity of the false-positive crossmatches was also highly variable, potentially due to donor- and/or cell-dependent expression of CD38. The variability of CD38 expression was, therefore, for the first time, characterized on the T and B cells isolated from various tissues and peripheral blood of 78 individuals. Overall, T cells were found to have a lower CD38 expression profile than the B cells, and no significant difference was observed between deceased and living individuals. Finally, we show that a simple cell treatment by dithiothreitol can effectively mitigate Daratumumab interference thus preserving the utility of pre-transplant crossmatch in multiple myeloma patients awaiting kidney transplant.

## 1. Introduction

Chronic kidney disease occurs in approximately 50% of multiple myeloma (MM) patients [1]. Historically, MM patients with end-stage renal disease (ESRD) were typically not considered for kidney transplant due to the overall poor survival rate of MM [2]. However, since the introduction of various novel therapeutics, the overall survival of MM patients has more than doubled [3]. Therefore, it is becoming more common practice for centers to transplant MM patients with ESRD. MM is characterized by cancerous plasma cells and is defined as incurable, but can be managed successfully in certain patients using new therapies [4]. Plasma cells express high levels of CD38. Therefore, monoclonal antibodies against CD38 have been developed to treat MM such as daratumumab (Dara) [5]. Dara is an IgG1κ human monoclonal antibody that binds to CD38 and inhibits the development of CD38-expressing cells through multiple mechanisms [6].

For renal transplant, flow cytometric crossmatch (FCXM) aids in the determination of alloimmune risk pre-transplant. Rejection occurs when antigens, mainly human leukocyte antigens (HLA), expressed on the donor allograft, activate the patient’s humoral immune response. A positive flow crossmatch due to donor-specific HLA antibodies is associated with worse allograft outcome and can be a contraindication to kidney transplant [7,8]. However, it is well-known that certain monoclonal antibody treatments can cause a false-positive FCXM (e.g., rituximab) [9,10]. Herein, we document the first case of Dara interference repeatedly causing aberrant false-positive FCXM, which inadvertently delayed a renal waitlisted MM patient from being transplanted. We also show that CD38 expression is highly variable on lymphocytes and can contribute to variability in FCXM positivity. Finally, we demonstrate a new application to alleviate Dara interference using a simple cell treatment while still preserving the utility of pre-transplant flow crossmatch in Dara-treated patients.

## 2. Materials and Methods

### 2.1. Lymphocyte Isolation

All lymphocytes were isolated from anti-coagulated peripheral blood, lymph nodes or spleen tissues via immunomagnetic negative selection using EasySepTM Direct Human Total Lymphocyte Isolation Kits and RoboSepTM-S (StemCell Technologies, Vancouver, BC, Canada) per the manufacturer’s instructions, followed by treatment with pronase (3.0 Kunitz units/mL) and DNase (deoxyribonuclease I; 2000 Kunitz units/mL) (Sigma-Aldrich, St. Louis, MO, USA) at 37 °C for 10 min.

### 2.2. Flow Cytometry

All crossmatches and expression experiments were performed on a BD FACSLyric flow cytometer and analyzed with FACSuite software (BD Biosciences, San Jose, CA, USA) in median channel shift (MCS) or median channel fluorescence (MCF). All monoclonal antibodies used for cell staining were obtained from BD Biosciences unless otherwise noted, which included APC-conjugated mouse anti-human CD45 for leukocytes (clone HI30), PerCP-conjugated mouse anti-human CD3 for T cells (clone SK7), and PE-conjugated mouse anti-human CD19 for B cells (clone SJ25C1). FITC-conjugated F(ab’)2 fragment goat anti-human IgG polyclonal antibody (Jackson ImmunoResearch, West Grove, PA, USA) was used for the detection of surface-bound donor-specific antibody. Cell surface CD38 expression was semi-quantitatively characterized using FITC-conjugated anti-human CD38 (clone HB7), while HLA expression was characterized by APC-conjugated anti-HLA class I (clone G46-2.6) and FITC-conjugated anti-HLA class II (clone Tu39). The lots of the secondary antibodies used, and their titrations, as applicable, were identical throughout the case study. To minimize run-to-run variability, all CD38 expression experiments were performed using a single lot of anti-CD38, performed by the same technologist, and on the same flow cytometer.

### 2.3. Dithiothreitol (DTT) Treatment

Working stock of DTT (Sigma-Aldrich) was reconstituted with Dulbecco’s phosphate-buffered saline and adjusted for pH to 7.0–7.5. Lymphocyte treatment with DTT was performed by incubating 1 × 10^6^ purified lymphocytes with 0.05 M or 0.1 M of DTT at 37 °C for 10, 20, or 30 min prior to pronase cell treatment. Treatment of serum with DTT was performed by incubating undiluted patient sera with 0.05 M DTT at 9:1 *v/v* ratio at 37 °C for 45 min.

### 2.4. Statistical Analysis

Statistical analyses were performed using student’s T test (Microsoft Excel, Redmond, WA, USA). *p* < 0.05 was considered statistically significant.

## 3. Results

The patient was a 67-year-old white female with ESRD secondary to MM. She was blood group AB and her sensitization included an autologous hematopoietic stem cell transplant over 20 years ago, multiple units of blood transfusions and two pregnancies. Solid phase testing using single antigen beads (One Lambda, West Hills, CA, USA) confirmed the absence of anti-HLA antibodies since registered on the renal waitlist (reactivities were all below 100 MFI; based on a positive threshold of 1000 MFI; Appendix A). However, the patient consistently had unexpected positive FCXM against the majority of the deceased donors offered (T cell and B cell positive, *n* = 21; T cell negative and B cell positive, *n* = 5; total *n* = 27; Appendix A). The strength of the positive FCXM as represented in MCS was highly variable (T cell, +45–158; B cell, +74–239; positive cutoff T cell > +39 MCS, B cell > +70 MCS; Appendix A). The presence of autoantibodies was subsequently ruled out by a negative autologous FCXM (Appendix A).

Upon further review and investigation, since the patient was being actively treated at an outside hospital for her MM with Dara (DARZALEX^®^; Janssen Biotech, Horsham, PA, USA) infusions, it was presumed that these positive T and B cell FCXM results were false-positive caused by Dara binding to donor cell surface CD38. However, very little information was available on CD38 expression on human T and B cells isolated from lymphoid tissues. Therefore, we sought to characterize CD38 expression in T cells and B cells isolated from various tissue types including peripheral blood, lymph nodes, and spleen from 71 deceased and 7 living donors (Figure 1). The CD38 expression level on donor lymphocytes, as represented and analyzed in MCF, was found to be highly variable and differed by as much as 83% (T cell, 238–428; B cell, 237–433; *n* = 78) (Figure. 1). Overall, T cells were found to have a lower CD38 expression profile than B cells (T cell = 296 ± 40 vs. B cell = 317 ± 38; *p* < 0.001), but no significant difference was observed between lymphocytes derived from deceased and living donors (T cell *p* = 0.15; B cell *p* = 0.59), although the number of living donors assessed for CD38 expression was conceivably underrepresented for definitive conclusion which may warrant further investigation. Conversely, CD38 expression on T cells isolated from lymph nodes was on average 16% higher than those from peripheral blood (lymph node = 325 ± 43 vs. peripheral blood = 281 ± 28; *p* < 0.0001), but CD38 expression on B cells was comparable regardless of tissue source (*p* = 0.10–0.54) (Figure 1).

### Elimination of False-Positive Reactivity

Aiming to mitigate interference from Dara on FCXM, we found that treatment of donor lymphocytes with a mild 0.05 M of DTT for 10 min at 37 °C prior to pronase treatment was able to cleave off the majority of the cell surface CD38 and effectively abrogate the false-positive FCXM caused by Dara binding (Table 1). Importantly, inferring from the lack of noticeable difference on the MCF values of the positive and negative control sera between untreated and DTT-treated lymphocytes in the two surrogate donor crosmatches, DTT cell treatment did not appear to affect the sensitivity or specificity of FCXM, or the level of cell surface HLA expression, although further experiments would be needed to confirm the findings. Increasing DTT concentration to 0.10 M or incubation time up to 30 min had no further impact on CD38 expression level and crossmatch outcomes (Appendix A). Treatment of serum with DTT alone was not effective in mitigating Dara interference (Appendix A).

## 4. Discussion

In this case study, we report the first case of Dara interference in routine FCXM in the setting of clinical solid organ transplantation. Furthermore, a simple and quick treatment of donor lymphocytes using DTT, which is a reagent readily available in most clinical histocompatibility laboratories, can effectively mitigate the interference from Dara and preserve the utility of FCXM. A confounding factor in this case was the variable T cell FCXM results. It has been reported that CD38 is highly expressed in MM cells and activated T cells, suggesting that CD38 cellular surface expression is highly dependent on tissue type, cell type, and level of cellular/tissue activation [4,11]. Notably, FCXM results are contingent on cellular properties, understanding of patient treatment, and stringent technique. These factors can impact expression of HLA molecules and other cell surface proteins—including CD38 [12,13]. Not surprisingly, B cell expression of CD38 was consistently detected despite a wide range in expression based on mean fluorescence intensity, while T cell expression of CD38 was also highly variable.

Previous studies in transfusion medicine have reported Dara-treated patients demonstrating panreactivity on red blood cell (RBC) panel testing [14]. Chapuy et al. [15] show the panreactivity by Dara-treated patients is by binding to CD38 expressed on RBCs and that treating RBCs with DTT removed Dara interference. However, the authors do note that DTT does denature Kell antigens demonstrating that one potential drawback of DTT treatment in removing Dara interference is the potential denaturing of certain antigens (e.g., HLA). In this study, we did not see a detrimental effect of DTT on HLA expression and crossmatch results, suggesting that HLA antigens are still intact. A limitation of this study was that we did not use soluble CD38 or anti-Dara to neutralize Dara in the serum. This was due to lack of access to Dara, cost, and lack of time to resolve the issue for this patient. However, based on previous elegant and comprehensive studies performed, as discussed above, treating cells with DTT is a robust method to remove Dara interference and is a plausible first step in addressing this issue. For the appearance of the rather limited reductions of cell surface CD38 expression after DTT lymphocyte treatment (up to 12% for T cell and 27% for B cell; Table 1), we reason that it was due to the use of MCF for semiquantitative assessments of the impact of DTT on CD38 expression, as opposed to sole qualitative examinations via the % change of CD38+ lymphocytes in the entire cell preparation.

One of the confounding issues associated with this case was the oversight that the patient was receiving Dara-infusions at an outside clinic. We do not have access to the exact dates of the infusions, but the patient started Dara infusions more than 6 months prior to the first serum draw for anti-HLA antibody testing and received monthly infusion thereafter. Studies have shown that following the end of Dara treatment, Dara was detectable in the majority of patients at 8 weeks post-treatment (reviewed in [16]). Concerning the one negative of the 27 deceased donor crossmatches, it was later discovered that, in addition to the possibility of variable CD38 expression, the negative crossmatch could, in part, be due to an analytical anomaly in the gating of the B cell population tagged with FITC-anti-IgG (in presence of distinct double peaks) and that the B cell was purportedly positive (while T cell was unequivocally negative). Future studies examining the direct inhibition of Dara are underway along with how the kinetics may impact results related to the amount of drug and CD38 expression but are outside of the scope of this case report.

In this case, the patient eventually received a deceased donor transplant with no complications pre- and post-transplant. However, there was significant post-transplant debate about the continuation of Dara treatment. After multidisciplinary input, hematology decided to continue with the current Dara treatment, as the immunological impact of ceasing treatment could be dramatic. Recently, Doberer et al. [17] proposed targeting CD38 in the treatment of chronic antibody-mediated rejection (AMR) due to the efficacy of Dara in ameliorating pathological features associated with chronic AMR in a kidney transplant recipient. Kwun et al. [18] provide experimental evidence that targeting CD38 with Dara significantly lowers the levels of anti-HLA antibodies in both human and non-human primates. Notably, when Dara treatment was stopped, all the recipients had a rebound in anti-HLA antibodies and T cell mediated rejection.

The immune system is a recursive matrix of regulation. Regulatory T cells and myeloid derived suppressor cells express CD38, as well, and likely are targeted by Dara—which may have unanticipated effects [19]. Taken together, there may be a viable approach for desensitization strategies using this drug but the discontinuation of this drug in the post-transplant setting, where it had been used for reasons other than humoral rejection, should include close monitoring for rejection due to the potential for a rebound memory response.

## Figures and Tables

**Figure 1 jcm-11-06059-f001:**
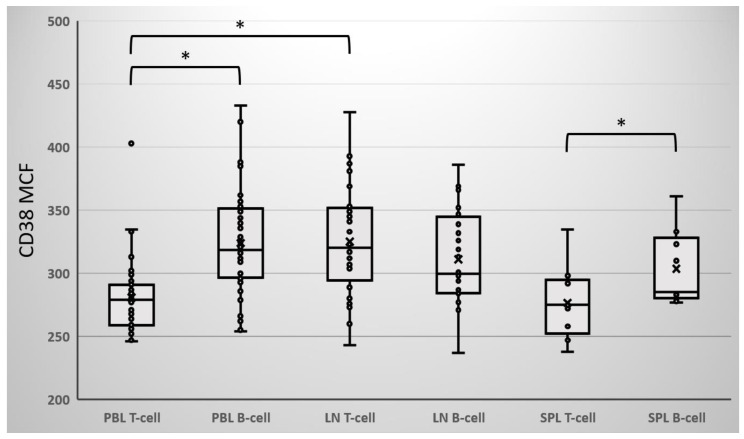
CD38 expression was highly variable depending on if T and B cells were isolated from peripheral blood (PBL), lymph nodes (LN), or spleen (SPL). The cell surface expression of CD38 was performed on pronase-treated T and B cells from 71 deceased and 7 living donors (*n* = 78). Crossmatches were performed on pronased donor lymphocytes using a BD FACSLyric flow cytometer with staining of CD45/leukocyte, CD3/T, CD19/B, G46-2.6/HLA-I and Tu39/HLA-II. Cell surface CD38 expression was characterized using HB7 anti-human CD38 monoclonal antibody and analyzed in median channel fluorescence (MCF) values. Student’s *t*-test, * *p* < 0.001.

**Table 1 jcm-11-06059-t001:** Donor lymphocyte treatment by 0.05 M of DTT for 10 min at 37 °C effectively removed surface-bound CD38 and abrogated false-positive flow cytometric crossmatch (FCXM) outcomes caused by patient sera containing Daratumumab without affecting assay sensitivity and specificity. Values for the control sera and the HLA class I, HLA class II and CD38 expression are represented and analyzed in median channel fluorescence (MCF). Values for the five patient sera used in the surrogate donor crossmatches (in parentheses) are represented in median channel shift (MCS) from the negative control (i.e., patient serum MCF minus negative control MCF). Crossmatch outcomes were interpreted from MCS based on established positive cutoffs. NA, not applicable.

	Surrogate Donor A				Surrogate Donor B			
	Neat (Untreated)		DTT-Treated Lymphocytes		Neat (Untreated)		DTT-Treated Lymphocytes	
	T	B	T	B	T	B	T	B
Positive Control Serum (Strong)	711	798	741 (↑4%)	817 (↑2%)	714	717	724 (↑1%)	724 (↑1%)
Positive Control Serum (Weak)	462	534	483 (↑5%)	555 (↑4%)	461	473	464 (↑1%)	465 (↓2%)
Negative Control Serum	297	294	302 (↑2%)	307 (↑4%)	281	258	315 (↑12%)	282 (↑9%)
HLA Class I Expression	481	529	509 (↑6%)	543 (↑3%)	560	550	600 (↑7%)	581 (↑6%)
HLA Class II Expression	NA	684	NA	677 (↓1%)	NA	584	NA	640 (↑10%)
CD38 Expression	258	260	226 (↓12%)	201 (↓23%)	246	254	227 (↓8%)	185 (↓27%)
Patient Serum 1	POS (42)	POS (79)	Neg (−8)	Neg (−57)	Neg (23)	POS (127)	Neg (−11)	Neg (−67)
Patient Serum 2	POS (42)	POS (86)	Neg (−14)	Neg (−67)	Neg (21)	POS (126)	Neg (−14)	Neg (−73)
Patient Serum 3	POS (42)	POS (91)	Neg (−18)	Neg (−63)	Neg (20)	POS (121)	Neg (−24)	Neg (−68)
Patient Serum 4	POS (41)	POS (83)	Neg (−15)	Neg (−63)	Neg (22)	POS (122)	Neg (−21)	Neg (−67)
Patient Serum 5	POS (42)	POS (85)	Neg (−15)	Neg (−59)	Neg (21)	POS (123)	Neg (−16)	Neg (−69)

## Data Availability

Not applicable.

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
