# Peer review of "Daratumumab Interferes with Allogeneic Crossmatch Impacting Immunological Assessment in Solid Organ Transplantation"

_jcm, 2022, doi:10.3390/jcm11206059_

Round 1
Reviewer 1 Report
20220929_Reviewer’s comment on Manuscript JCM ID jcm1937219
The manuscript of Ho CS et al. presents a Case Study illustrating interference of Daratumumab with allogenic crossmatch test results. The description of this fact, as well as the procedure proposed by the study authors to eliminate this problem, are important contributions to improve Flow-XM result interpretation in the context of pre-transplant risk assessment. The manuscript is well written, with some details mainly missing in the Methods, the Results are clear and the Discussion Chapter comprehensively processes the case in the overall context of the topic. Most of the identified flaws are minor, and are addressed below:
1_Title: the authors are using the term “histocompatibility crossmatch”, however, this is misleading: a crossmatch assay, most commonly performed by incubating donor cells with patient serum, examines, if donor-specific antibodies are present or not. Thus, the crossmatch test has nothing to do with the (degree of) histocompatibility (histocompatibility is differently tested, namely by donor and recipient HLA typing and subsequent comparison of the results). Therefore, another title is required, without the term “histocompatibility” (e.g. “Daratumumab interferes with allogeneic crossmatch-tests impacting immunological risk assessment in solid organ transplantation”).
2_Reference 7: the authors refer to a study NOT including Flow-crossmatch results. The same group, however, has realized a similar study INCLUDING Flow-XM results:
Bielmann D, Hönger G, Lutz D, Mihatsch MJ, Steiger J, Schaub S. Pretransplant risk assessment in renal allograft recipients using virtual crossmatching. Am J Transplant. 2007 Mar;7(3):626-32. doi: 10.1111/j.1600-6143.2007.01667.x. PMID: 17352712.
This Reference would support the statement of line 50 and 51 more accurately.
3_Reference 9: the authors have chosen a review which only mentions the topic (interference of rituximab with Flow XM results) superficially. It would be more appropriate, to refer to original publications which are fully subject related, e.g.:
Desoutter J, Apithy MJ, Bartczak S, Guillaume N. False Positive B-Cells Crossmatch after Prior Rituximab Exposure of the Kidney Donor. Case Rep Transplant. 2016;2016:4534898. doi: 10.1155/2016/4534898. Epub 2016 Apr 28. PMID: 27239362; PMCID: PMC4864553.
Doss SA, Mittal S, Daniel D. Impact of rituximab on the T-cell flow cytometric crossmatch. Transpl Immunol. 2021 Feb;64:101360. doi: 10.1016/j.trim.2020.101360. Epub 2020 Dec 22. PMID: 33359130.
4_The sentence in line 71-75 reads difficult. I think the authors should use a semicolon or “such as” or another linker between the bracket closed sign and the abbreviation APC.
5_Line 95-96: it would be valuable if the authors would additionally indicate the applied cutoff MFI (for the SAB assay).
6_Methods_Flow Cytometry: The authors have quantitatively assessed HLA class I and II expression as well as CD38 expression using FITC-conjugated secondary antibodies. It is important that they explain in more details what standardization they applied to ensure that measured differences (results) is due to the real (biological) difference in expression, and NOT due to technical variation. The two main sources of technical variability may come from
A_inter-assay variation: I assume that it was not possible to perform mentioned expression measurements (with peripheral white blood cells, lymph nodes and spleen cells from 71 deceased donors) within the same experiment. How did the authors standardize their different test runs (calibration, using same controls, normalizations?)
B_using different lots of 2nd Ab: it is known that conjugation of antibody with small fluorochromes (such as FITC) will reveal different fluorochrome-protein ratio. Indeed, each lot can represent another ratio. The lot specific ratio is either printed on the lot specific product sheet or can be requested by the product provider. If the authors have indeed used different Lots of their 2nd (FITC-conjugated) 2nd Abs: did they normalize their results for the variable fluorochrome-protein ratio?
The actions taken by the authors to prevent such additional impacts should be indicated in the Methods section. And, since at least one of the two mentioned influences may have probably impacted the results (e.g. line 111: “was found to be highly variable”), the possible impacts on the results should be explained in the Discussion Chapter and mentioned as a limitation.
7_in line 125, the authors state: “Importantly, DTT cell treatment did not affect the sensitivity or specificity of FCXM, or the level of cell surface HLA expression”… To support this statement, it is necessary that the authors compare their measured difference (untreated versus DTT treated) with their measured NATURAL variability (e.g. MCF values from multiple measurements of negative control with the very same T- (or B-) cells). Only if DTT treatment reveals a similar (or lower) variability than the observed normal variability, the statement “did not affect sensitivity or specificity of FCXM” can be made. In other words and as an example: if the normal variability of MCF values from the negative control serum with T cells from surrogate donor B (Table 1) would be 7% (e.g.), your measured change of 12% (after DTT-treatment) would be significantly higher, thus your statement would not be applicable.
8_Table 1: although it is understandable that the authors want to primarily show to the reader that DTT treatment did change the FCXM result from POS to neg, it would be more informative to indicate numeric (MCF) values (in the five rows showing results of patient serum 1 to 5). A color code for POS and neg could still be applied (or (POS) and (neg) could be indicated in brackets after the value. I think the reader are curious about how much the MCF values are dropping, respectively how much they are dropping below the values of the negative control serum.
9_Supplementary Table 3: the authors are displaying MCF and MCS results within the same table. It would be helpful to better indicate, which of the two is applied (e.g. “Positive Control Shift (weak)” or “HLA class I expression (MCF)”, or “HLA class I” expression (MCF)”, or “CD38 expression (MCF)”. Furthermore, to me it is not clear, if the (negative) values (e.g. “-8”) displayed in the DTT (cell) treatment columns are delta-values (difference between pre- and post-treatment) or not.
Author Response
Response to Reviewer Comments
Reviewer 1
The manuscript of Ho CS et al. presents a Case Study illustrating interference of Daratumumab with allogenic crossmatch test results. The description of this fact, as well as the procedure proposed by the study authors to eliminate this problem, are important contributions to improve Flow-XM result interpretation in the context of pre-transplant risk assessment. The manuscript is well written, with some details mainly missing in the Methods, the Results are clear and the Discussion Chapter comprehensively processes the case in the overall context of the topic.
Most of the identified flaws are minor, and are addressed below:
1_Title: the authors are using the term “histocompatibility crossmatch”, however, this is misleading: a crossmatch assay, most commonly performed by incubating donor cells with patient serum, examines, if donor-specific antibodies are present or not. Thus, the crossmatch test has nothing to do with the (degree of) histocompatibility (histocompatibility is differently tested, namely by donor and recipient HLA typing and subsequent comparison of the results). Therefore, another title is required, without the term “histocompatibility” (e.g. “Daratumumab interferes with allogeneic crossmatch-tests impacting
immunological risk assessment in solid organ transplantation”).
Authors’ Response: We thank the reviewer for the suggestion. The title has been updated with the removal of “histocompatibility”.
2_Reference 7: the authors refer to a study NOT including Flow-crossmatch results. The same group, however, has realized a similar study INCLUDING Flow-XM results:
Biermann D, Hanger G, Lutz D, Martsch MJ, Setiger J, Schaub Pretransplant risk assessment in renal allograft recipients using virtual crossmatching. Am J Transplant. 2007 Mar;7(3):626-32. doi:10.1111/j.1600-6143.2007.01667.x. PMID: 17352712.
This Reference would support the statement of line 50 and 51 more accurately.
Authors’ Response: We thank the reviewer for this feedback. The references have been updated.
3_Reference 9: the authors have chosen a review which only mentions the topic (interference of rituximab with Flow XM results) superficially. It would be more appropriate, to refer to original publications which are fully subject related, e.g.:
Declutter J, Apathy MJ, Bartizan S, Guillaume N. False Positive B-Cells Crossmatch after Prior Rituximab Exposure of the Kidney Donor. Case Rep Transplant. 2016;2016:4534898. doi:10.1155/2016/4534898. Epub 2016 Apr 28. PMID: 27239362; PMCID: PMC4864553.
Doss SA, Mittal S, Daniel D. Impact of rituximab on the T-cell flowcytometric crossmatch. Transept Immunol. 2021 Feb;64:101360. doi: 10.1016/j.trim.2020.101360. Pub 2020 Dec 22. PMID:33359130.
Authors’ Response: We thank the reviewer for this feedback. The references have been updated.
4_The sentence in line 71-75 reads difficult. I think the authors should use a semicolon or “such as” or another linker between the bracket closed sign and the abbreviation APC.
Authors’ Response: We thank the reviewer for this feedback. The sentence has been revised to improve the read and clarity.
5_Line 95-96: it would be valuable if the authors would additionally indicate the applied cutoff MFI (for the SAB assay).
Authors’ Response: We thank the reviewer for this feedback. The positive MFI cutoff for the SAB assays has been added, and representative SAB histograms have been added as supplementary Figure S1.
6_Methods_Flow Cytometry: The authors have quantitatively assessed HLA class I and II expression as well as CD38 expression using FITC-conjugated secondary antibodies. It is important that they explain in more details what standardization they applied to ensure that measured differences (results) is due to the real (biological) difference in expression, and NOT due to technical variation. The two main sources of technical variability may come from A_inter-assay variation: I assume that it was not possible to perform mentioned expression measurements (with peripheral white blood cells, lymph nodes and spleen cells from 71 deceased donors) within the same experiment. How did the authors standardize their different test runs (calibration, using same controls, normalizations?)
B_using different lots of 2nd Ab: it is known that conjugation of antibody with small fluorochromes (such as FITC) will reveal different fluorochrome-protein ratio. Indeed, each lot can represent another ratio.
The lot specific ratio is either printed on the lot specific product sheet or can be requested by the product provider. If the authors have indeed used different Lots of their 2nd (FITC-conjugated) 2nd Abs: did they normalize their results for the variable fluorochrome-protein ratio?
The actions taken by the authors to prevent such additional impacts should be indicated in the Methods section. And, since at least one of the two mentioned influences may have probably impacted the results (e.g. line 111: “was found to be highly variable”), the possible impacts on the results should be explained in the Discussion Chapter and mentioned as a limitation.
Authors’ Response: We thank the reviewer for the questions and feedback. We did not experience any change to the antibody lots or titrations throughout the case. To minimize run-to-run variability, all CD38 expression experiments were performed using a single lot of anti-CD38, by the same technologist, and on the same flow cytometer. Additional clarifications have been added to Section
2.2 “Flow Cytometry”.
7_in line 125, the authors state: “Importantly, DTT cell treatment did not affect the sensitivity or specificity of FCXM, or the level of cell surface HLA expression”... To support this statement, it is necessary that the authors compare their measured difference (untreated versus DTT treated) with their measured NATURAL variability (e.g. MCF values from multiple measurements of negative control with the very same T- (or B-) cells). Only if DTT treatment reveals a similar (or lower) variability than the observed normal variability, the statement “did not affect sensitivity or specificity of FCXM” can be made. In other words and as an example: if the normal variability of MCF values from the negative control serum with T cells from surrogate donor B (Table 1) would be 7% (e.g.), your measured change of 12% (after DTT-treatment) would be significantly higher, thus your statement would not be applicable.
Authors’ Response: We thank the reviewer for the feedback. The statement was made, in more of a qualitative manner, based on the observations in the 2 surrogate crossmatches that there was no meaningful difference between untreated and DTT-treated cells on the MCF of both the strong and weak positive controls (i.e. sensitivity) as well as the negative controls (i.e. specificity) (though, only the values of 1 of the 4 negative controls used was shown in Table 1 to minimize confusions).
Additional clarifications have been added to Section 3.1.
8_Table 1: although it is understandable that the authors want to primarily show to the reader that DTT treatment did change the FCXM result from POS to neg, it would be more informative to indicate numeric (MCF) values (in the five rows showing results of patient serum 1 to 5). A color code for POS and neg could still be applied (or (POS) and (neg) could be indicated in brackets after the value. I think the reader are curious about how much the MCF values are dropping, respectively how much they are dropping below the values of the negative control serum.
Authors’ Response: We thank the reviewer for the suggestion. Table 1 has been revised to include the patient sera MCS values.
9_Supplementary Table 3: the authors are displaying MCF and MCS results within the same table. It would be helpful to better indicate, which of the two is applied (e.g. “Positive Control Shift (weak)” or “HLA class I expression (MCF)”, or “HLA class I” expression (MCF)”, or “CD38 expression (MCF)”.
Furthermore, to me it is not clear, if the (negative) values (e.g. “-8”) displayed in the DTT (cell) treatment columns are delta-values (difference between pre- and post-treatment) or not.
Authors’ Response: We thank the reviewer for the feedback. To minimize confusion, Table S3 hasvbeen revised to include only the MCS values for the crossmatch sera. The MCF for the positive/negative controls and HLA and CD38 expression are already included in Table 1.
Reviewer 2 Report
In “Daratumumab interferes with histocompatibility crossmatch impacting immunological assessment in solid organ transplantation”, Ho et al. present the first purported case of Daratumumab interference of allogeneic crossmatch in the setting of solid organ transplantation. They ascribe false-positive crossmatch results to Daratumumab, a claim supported by the absence of HLA antibodies on single antigen bead assay and the abrogation of reactivity with DTT, which can decrease expression of CD38, the target of Daratumumab. Furthermore, the authors demonstrated variable CD38 expression on a number of deceased donor and living donor lymphocytes using a different CD38 antibody. The case report is well written and the experiments are well designed and described.
Rituximab, IVIG and ATG are well known to interfere with HLA antibody testing including crossmatching. However, Daratumumab has not been considered as a possible interference. As such, the findings in this paper would appear novel and are very important in regards to evaluating recipient:donor histocompatibility pre-transplantation. As the present case demonstrates, positive crossmatches may delay transplantation, which prolongs wait times on the transplant list and the time on dialysis, both of which have been shown to adversely affect outcomes post-transplantation. The transplant community, in particular HLA directors and laboratorians, would be the ideal audience who would find the findings herein of the most interest and clinically applicable.
This reviewer would recommend acceptance pending revisions. The following are comments are for the authors’ consideration:
- Can the authors describe the timing of the patient’s last Daratumumab infusion compared to when the sera samples used in the 27 crossmatches? Were crossmatches performed closer to infusions more reactive than those using sera samples drawn weeks after infusion? Time of infusion to sera sample could be an important consideration when trying to interpret results in similar patients.
- The authors say 26/27 crossmatches performed with cells from deceased donors were positive. Concerning the one that was negative, were these cells examined? Was there less CD38 expression on these cells compared to others, potentially explaining the negative reaction? Furthermore, were these 27 deceased donors included in the 71 deceased donors assessed for CD38 expression? If so, it would be prudent to correlate the expression with the crossmatch reactivity.
- The authors provide comparison between CD38 expression on deceased donor and living donor cells but the difference in number is considerable with 71 deceased donors and 7 living donors. Seven living donors may not provide an accurate assessment for living donors as a whole. Considering published data demonstrating differential levels of HLA expression between deceased donor and living donor cells, it would support the authors’ claims to increase the number of living donors evaluated.
- The authors mention that an autologous crossmatch was performed but data was not supplied. Was this performed more than once and by more than one technologist? If only performed once and the reactivity was close to the positive cutoff, it could easily be positive performed by another technologist. Multiple myeloma is a disease in which there is overproduction of immunoglobulins, which are known to cause issues in serologic testing such as the direct anti-globulin test and reverse typing in the blood bank. Thus, is it possible that some reactivity seen in crossmatches is due to non-specific binding of the immunoglobulins in the patient’s plasma? Do this end, perhaps the authors could supply results of autologous crossmatch, and if not already done, have repeats performed by different technologists. Additionally, it may be helpful to provide the patient’s immunoglobulin levels at the time of testing. If they are low, then this would argue against false positivity due to the patient’s disease process.
- The authors state that the HLA antibody testing by single antigen bead analysis was negative. In this patient with a history of sensitization (transfusion and pregnancy), did she ever have antibody? Could it be the Daratumumab decreased antibody levels? Also, were the single antigen bead assays completely negative or was there antibody below the positive threshold? It may be possible that there is a low level antibody to an epitope shared amongst many beads thus imparting lower MFI values. It has been reported that such antibodies with low MFI values can still correlate with positive crossmatches. Therefore, it would be judicious to share the laboratory’s cutoff for assigning antibody specificity and report if there was any low level antibody below this threshold.
- In table 1, the CD38 expression between the Neat and DTT-treated columns does not seem to be terribly different. Is this due to the fact that MCFs were used and small numerical differences represent larger shifts than reflected by the numerical change? Please offer some explanation in the discussion, as this may be a point of clarification needed for the readership.
Author Response
Response to Reviewer CommentsReviewer 2
In “Daratumumab interferes with histocompatibility crossmatch impacting immunological assessment in solid organ transplantation”, Ho et al. present the first purported case of Daratumumab interference of allogeneic crossmatch in the setting of solid organ transplantation. They ascribe false-positive
crossmatch results to Daratumumab, a claim supported by the absence of HLA antibodies on single antigen bead assay and the abrogation of reactivity with DTT, which can decrease expression of CD38, the target of Daratumumab. Furthermore, the authors demonstrated variable CD38 expression on a number of deceased donor and living donor lymphocytes using a different CD38 antibody. The case report is well written and the experiments are well designed and described.
Rituximab, IVIG and ATG are well known to interfere with HLA antibody testing including crossmatching. However, Daratumumab has not been considered as a possible interference. As such, the findings in this paper would appear novel and are very important in regards to evaluating recipient: donor histocompatibility pre-transplantation. As the present case demonstrates, positive crossmatches may delay transplantation, which prolongs wait times on the transplant list and the time on dialysis, both of which have been shown to adversely affect outcomes post-transplantation. The transplant community,
in particular HLA directors and laboratorians, would be the ideal audience who would find the findings herein of the most interest and clinically applicable. This reviewer would recommend acceptance pending revisions. The following are comments are for the authors’ consideration:
- Can the authors describe the timing of the patient’s last Daratumumab infusion compared to when the sera samples used in the 27 crossmatches? Were crossmatches performed closer to infusions more reactive than those using sera samples drawn weeks after infusion? Time of infusion to sera sample could be an important consideration when trying to interpret results in similar patients. The authors say 26/27 crossmatches performed with cells from deceased donors were positive. Concerning the one that was negative, were these cells examined? Was there less CD38 expression on these cells compared to others, potentially explaining the negative reaction? Furthermore, were these 27 deceased donors included in the 71 deceased donors assessed for CD38 expression? If so, it would be prudent to correlate
the expression with the crossmatch reactivity.
Authors’ Response: We thank the reviewer for bringing up this important point and we agree that the timing of Daratumumab infusion and potential impact on the FCXM is important. Unfortunately, we do not have the exact dates of when the patient received infusions but we do know when treatment was initiated and that they were treated on a monthly basis. We have added in this into discussion starting on line 198.
Regarding the only one negative crossmatch, it was later discovered that, while T cell was unequivocally negative, two distinct B cell FITC peaks were detected in that donor and that only the more negative peak (to the left in the B cell FITC histogram) was gated for MCS analysis (excluding the more positive peak to the right). Additional explanation for the negative crossmatch has been added.
On the other hand, the intent to characterize CD38 cell surface expression on the lymphocytes from 71 individuals was in fact evolved from the observation of the variable MCS with the unexpected positive crossmatches and hence the hypothesis that CD38 expression may also be highly variable on T and B cells, which would then, in part, account for the variable MCS. The 71 deceased/living assessed for CD38 expression were independent from the initial 27 deceased donors, and therefore, we were unable to correlate the variability of CD38 expression with the variability of the MCS.
- The authors provide comparison between CD38 expression on deceased donor and living donor cells but the difference in number is considerable with 71 deceased donors and 7 living donors. Seven living donors may not provide an accurate assessment for living donors as a whole. Considering published data demonstrating differential levels of HLA expression between deceased donor and living donor cells, it would support the authors’ claims to increase the number of living donors evaluated.
Authors’ Response: We thank the reviewer for the feedback and agree that that the number of living donors assessed for CD38 expression may have been underrepresented for reliable conclusion. Additional clarifications have been added to the Results section.
- The authors mention that an autologous crossmatch was performed but data was not supplied. Was this performed more than once and by more than one technologist? If only performed once and the reactivity was close to the positive cutoff, it could easily be positive performed by another technologist.
Multiple myeloma is a disease in which there is overproduction of immunoglobulins, which are known to cause issues in serologic testing such as the direct anti-globulin test and reverse typing in the blood bank. Thus, is it possible that some reactivity seen in crossmatches is due to non-specific binding of the immunoglobulins in the patient’s plasma? Do this end, perhaps the authors could supply results of autologous crossmatch, and if not already done, have repeats performed by different technologists.
Additionally, it may be helpful to provide the patient’s immunoglobulin levels at the time of testing. If they are low, then this would argue against false positivity due to the patient’s disease process.
Authors’ Response: We thank the reviewer for the feedback and comments. The autologous flow crossmatch, which had only been performed once during the case, has been added as a
supplementary Table S4. Although one could interpret that the auto T cell MCS were elevated (but still unequivocally below the established cutoff) and could have contributed to the positivity observed in the deceased donor crossmatches, the magnitude of the MCS did not appear to be strong enough to account for the MCS observed in some of the allo crossmatches (Table S1; up to 158 T-MCS). Also, several of the patient sera had been used repeatedly in different deceased donor crossmatches. We
would expect to see more consistency in MCS if the positivity was largely contributed by the patient’s own disease process and immunoglobulin levels. Unfortunately, we did not have access to the patient’s immunoglobin levels.
The authors state that the HLA antibody testing by single antigen bead analysis was negative. In this patient with a history of sensitization (transfusion and pregnancy), did she ever have antibody? Could it be the Daratumumab decreased antibody levels? Also, were the single antigen bead assays completely
negative or was there antibody below the positive threshold? It may be possible that there is a low level antibody to an epitope shared amongst many beads thus imparting lower MFI values. It has been reported that such antibodies with low MFI values can still correlate with positive crossmatches. Therefore, it would be judicious to share the laboratory’s cutoff for assigning antibody specificity and report if there was any low level antibody below this threshold.
Authors’ Response: We appreciate the reviewer’s pointing out some of the important limitations of the SAB assay system related to shared epitopes (“peanut butter effect”) and/or prozone-like
inhibition, and these issues were thoroughly scrutinized in this case. Even though the patient has a history of allo sensitization (i.e. 2 pregnancies, multiple transfusions and an autologous HSC
transplant >20 years ago) as detailed at the beginning of the section 3 “Results”, she never had HLA antibody since waitlisted as routinely monitored for antibody by SAB. Representative SAB histograms have been added as supplementary Figure S1 to show a clear absence of HLA antibodies (all reactivities <100 MFI). The positive MFI cutoff for the SAB assays (1000 MFI) has also been added.
In table 1, the CD38 expression between the Neat and DTT-treated columns does not seem to be terribly different. Is this due to the fact that MCFs were used and small numerical differences represent larger shifts than reflected by the numerical change? Please offer some explanation in the discussion, as this
may be a point of clarification needed for the readership.
Authors’ Response: We thank the reviewer for the feedback. It is indeed logical to reason the “limited” reductions in MCF after DTT treatment were due to the fact that MCFs of the untreated
lymphocytes, as a whole, were used as the baseline fluorescence for semiquantitatively calculating the change in cell surface CD38 expression (as expressed in % change), as opposed to the absolute change in CD38 fluorescence. An alternative to this semiquantitative assessment was the use of a different gating strategy looking for more of a qualitative shift of the % of CD38+ cells out of a preestablished CD38+ gate after DTT treatment (e.g. pre-DTT, 80% T cells were CD38+; post-DTT, 0% T
cells were CD38+). We elected to measure and analyze using MCF with the intent to obtain a more (semi)quantitative assessment for finetuning the DTT treatment parameters (e.g. concentration and incubation time). Additional clarifications have been added to the Discussion section.